# Biofilm-Forming Ability of Phytopathogenic Bacteria: A Review of its Involvement in Plant Stress

**DOI:** 10.3390/plants12112207

**Published:** 2023-06-03

**Authors:** María Evangelina Carezzano, María Fernanda Paletti Rovey, Lorena del Rosario Cappellari, Lucas Antonio Gallarato, Pablo Bogino, María de las Mercedes Oliva, Walter Giordano

**Affiliations:** 1Instituto de Biotecnología Ambiental y Salud (INBIAS-CONICET), Córdoba X5804BYA, Argentina; ecarezzano@exa.unrc.edu.ar (M.E.C.); mpalettirovey@exa.unrc.edu.ar (M.F.P.R.); lcappellari@exa.unrc.edu.ar (L.d.R.C.); lgallarato@exa.unrc.edu.ar (L.A.G.); pbogino@exa.unrc.edu.ar (P.B.); moliva@exa.unrc.edu.ar (M.d.l.M.O.); 2Departamento de Biología Molecular; Universidad Nacional de Río Cuarto (UNRC), Río Cuarto, Córdoba X5804BYA, Argentina; 3Departamento de Microbiología e Inmunología, UNRC, Río Cuarto, Córdoba X5804BYA, Argentina

**Keywords:** biotic stress, biofilm, phytopathogenic bacteria, bacteria-plant interactions

## Abstract

Phytopathogenic bacteria not only affect crop yield and quality but also the environment. Understanding the mechanisms involved in their survival is essential to develop new strategies to control plant disease. One such mechanism is the formation of biofilms; i.e., microbial communities within a three-dimensional structure that offers adaptive advantages, such as protection against unfavorable environmental conditions. Biofilm-producing phytopathogenic bacteria are difficult to manage. They colonize the intercellular spaces and the vascular system of the host plants and cause a wide range of symptoms such as necrosis, wilting, leaf spots, blight, soft rot, and hyperplasia. This review summarizes up-to-date information about saline and drought stress in plants (abiotic stress) and then goes on to focus on the biotic stress produced by biofilm-forming phytopathogenic bacteria, which are responsible for serious disease in many crops. Their characteristics, pathogenesis, virulence factors, systems of cellular communication, and the molecules implicated in the regulation of these processes are all covered.

## 1. Introduction

Despite the many definitions of the term “stress” in plant physiology, there is a consensus that it is any biotic or abiotic environmental factor that reduces the rate of a physiological process, such as growth or photosynthesis, below a maximum value that would be attained under optimal conditions [1]. Plant survival under stress depends on the ability to perceive the stressful stimulus, generate and transmit appropriate signals, and initiate genetic, molecular, and physiological changes to cope with it.

Depending on the causal agent, abiotic stress (i.e., the stress produced by non-living elements) can be either physical or chemical. A deficit or an excess of water, high or low temperatures, salinity, UV radiation, and mechanical forces such as the wind are all physical sources of stress. Among the chemical factors, exposure to heavy metals and the lack of mineral nutrients stand out [2,3]. On the other hand, biotic stress is produced by other living organisms, such as bacteria, fungi, viruses, and insects.

Most plants are exposed to different types of stress at some point in their development. The effects include alterations in growth, yield, and quality as well as damage to cells, tissues, and organs. Drought and salinity, for instance, result in osmotic stress, which inhibits growth and causes metabolic disturbances [4]. Differentiating between the effects of abiotic and biotic stress can be difficult: some of the symptoms brought about by drought or a fungal infection, for example, may be similar. Whatever the cause, plant stress negatively impacts agriculture and is, therefore, a threat to global food production. Biotic factors, in particular, are devastating for economically important crops and for the environments in which those crops grow [5,6].

## 2. Abiotic Stress in Plants and Its Relation to Pathogenesis

### 2.1. Salt Stress

Soil salinization is one of the most limiting abiotic factors for agricultural productivity. It affects the physical-chemical properties of the soil (texture, osmotic potential, porosity, water conductivity, and aeration), increases the water holding capacity and osmotic pressure, and modifies the ecological balance in cultivated areas [7]. Unlike drought or heat stress, which are often intermittent and either precede or follow pathogen infection, salt stress tends to persist throughout most of the plant-growth cycle. This means that it might occur at the same time as other kinds of abiotic or biotic stress [8].

About 20% of farming land worldwide is affected by salinity [9], and this percentage is expected to reach about 50% by 2050. Most of the world’s saline soils are found in arid or semi-arid climates [10]. The problem is compounded by climate change everywhere, but it is especially serious for crops that require irrigation due to the lack of water in the areas where they grow. The salt in irrigation water is likely to accumulate in the soil, regardless of whether the water is used by the plants or lost to evaporation. The improper application of fertilizers and industrial pollution also contribute to the issue [11].

Soil salinity occurs when the soil’s water potential is reduced by cations, such as Na^+^ (sodium), Ca^+2^ (calcium), and K^+^ (potassium), and anions, such as Cl^−^ (chloride) and NO_3_^−^ (nitrate) [12]. This makes it difficult for plants to uptake water and nutrients and results in osmotic stress. In addition, certain saline ions may be absorbed and accumulated in plant tissue at concentrations that can become toxic and provoke physiological disorders. Especially high concentrations can modify the absorption of essential nutrients, upset the normal nutritional balance, and cause necrosis and premature death in older leaves [13,14]. Visible signs of exposure to salinity also comprise decreased seed germination, growth, development, flowering, and fruiting [15].

Early plant responses to salt stress, which are relatively well understood, include changes in free cytoplasmic Ca^2+^, the activation of Ca^2+^/calmodulin-dependent kinase, and the production of secondary signaling molecules such as reactive oxygen species (ROS) [14]. ROS lead to the deterioration of photosynthetic pigments, lipid peroxidation, alterations in the selective permeability of the cell membrane, protein denaturation, and DNA mutations [16,17]. Nevertheless, plants have protection and repair systems to mitigate this damage, and some species have evolved protective mechanisms featuring enzymatic and non-enzymatic components [18].

The response and adaptation to salt stress require the integration and coordination of multiple phytohormones, e.g., abscisic acid (ABA), jasmonic acid (JA), gibberellic acid (GA), ethylene, and salicylic acid (SA). ABA’s involvement is particularly significant in the response to abiotic stress. In fact, osmotic stress in roots leads to a very rapid and massive increase in the concentration of ABA, in both root and leaf tissues.

Although salt stress often takes place simultaneously with biotic stress, as mentioned earlier, few studies have focused on plant responses to these combinations of stimuli. Many abiotic stress conditions weaken plant defense mechanisms and thus enhance susceptibility to pathogen infection [19,20]. For example, two studies found associations between increased soil salinity and increased vulnerability to soil-borne diseases and *Phytophthora* spp. in tomatoes (*Solanum lycopersicum*) [21,22]. In both cases, saline stress was produced with different concentrations of sodium chloride (NaCl). Di Leo et al. also used calcium [22]. However, not all the interactions between biotic and abiotic sources of stress are harmful to plants. The nature of these interactions often depends on the timing, type, and severity of each stress involved. In barley, an increase in salt-induced (NaCl) osmotic stress was directly correlated with resistance to powdery mildew [23].

### 2.2. Drought Stress

Drought stress occurs when the plant’s water demand exceeds the amount available in the soil at a certain time, which creates an imbalance between water loss through transpiration and water uptake [24,25]. Ordinarily quite destructive, this type of stress has dramatically increased in intensity over the past few decades because of climate change [9]. The proportion of the Earth’s surface suffering from water deficit is expected to increase in the future, due to anthropogenic pollution and environmental degradation [26].

Since drought affects the uptake of essential nutrients from the soil, there is a subsequent decrease in the photosynthetic rate and thus in the availability of carbon (CO_2_). Abnormal plant growth and development ensue: the leaves are smaller, the stems are more elongated, and the roots highly proliferated [27]. Low water availability, moreover, disrupts cellular homeostasis [28] and triggers an increase in ROS production in different cell compartments. Low ROS levels are typically necessary for the normal progression of several biological processes, and the molecules can participate as second messengers in stem-cell maintenance, division, and differentiation as well as in organogenesis [29]. Failure to bring high intracellular ROS concentrations under control undermines the cell structure due to lipid peroxidation, protein oxidation, nucleic acid damage, enzyme inhibition, and the activation of programmed cell death pathways [4,28]. To prevent this, plants have enzymatic and non-enzymatic antioxidant systems [30]. The first consists of enzymes such as superoxide dismutase (SOD), catalase (CAT), peroxidase (PX), glutathione reductase (GR), and ascorbate peroxidase (APX). The second involves reducing agents such as phenolic compounds, glutathione, and aromatic amines [30,31].

When it comes to the interactions between drought stress and biotic stress and their impact on plants, examples are available but scarce in the literature. A study by Achuo et al. (2006) [32] found that drought stress reduced susceptibility to powdery mildew and *Botrytis cinerea* at the same time that the ABA concentration increased in tomatoes. Prasch and Sonnewald (2013) [33] investigated the response of *Arabidopsis* plants to stress caused by heat, drought, viral infection, and their combinations. The significant reduction in biomass observed with every source of stress on its own was exacerbated when they were acting together. Stomata closed upon exposure to viral infection and drought, viral infection, and heat, and all three simultaneously. Instead, heat stress or virus infection alone resulted in stomatal opening. Sorghum and common bean plants subjected to drought were more susceptible to *Macrophomina phaseolina,* the charcoal rot fungus [34,35]. Similarly, drought stress increased the spread of fungal and bacterial leaf scorch symptoms in date palms and the *Parthenocissus quinquefolia* vine [36,37].

## 3. Plant Resistance to Biotic Stress and Survival/Infection Strategies by Phytopathogenic Bacteria

Plants can be exposed to multiple sources of environmental stress at any given time, so the response to one stressful variable can negatively influence the ability to respond to another. In general, the physiological modifications that plants undergo when facing abiotic stress make them more susceptible to biotic stress. For instance, prolonged drought weakens development and increases vulnerability to attacks by pathogens. Moreover, certain bacteria can thrive in abiotic conditions that are unfavorable for plants. This is the case of biofilm formation in high-salinity soils, which has been reported to enhance cell viability in pathogenic bacteria and thus the likelihood of plant disease [38,39,40].

The control of diseases caused by phytopathogenic bacteria would benefit from a deeper understanding of how these bacteria survive within the host and of how plants respond to them [41]. Bacteria can come into contact with the host plant in different ways. One of them is chemotaxis, i.e., they can be attracted by substances that the plant synthesizes. They may also live as epiphytes on the plant surface until conditions are optimal for infection. Alternatively, they may be spread by infected seeds, transplants, tubers, insect vectors, machinery, wind, water, weeds, and stubble [42,43].

Plants have evolved a series of defense barriers against pathogenic bacteria. The first one is physical and passive: it is made up of trichomes, waxes, and cuticles in the epidermis, which makes it difficult for pathogens to establish themselves. The next line of defense consists of antimicrobial compounds and secondary metabolites, and it is activated when the plant’s immune system recognizes a pathogenic microorganism. This recognition takes place through a two-level signaling system. In the first level, pathogen-associated molecular patterns (PAMPs), such as flagellin and lipopolysaccharides (LPS), are sensed by proteins known as PAMP recognition receptors (PPR). In the second level, other proteins or intracellular immune receptors called NLRs (nucleotide-binding leucine-rich repeat proteins) recognize pathogen effectors. As a result of these two levels coming into play, two immune responses are possible. One of them, SAR (systemic acquired resistance), is made possible thanks to the synthesis of SA (salicylic acid). Commonly deployed against biotrophic and hemibiotrophic pathogens, SAR is hypersensitive and localized and, in the long term, confers protection to the entire plant against subsequent infections. During SAR, programmed cell death pathways are activated for diseased cells and their surroundings. Pathogen distribution throughout the plant is thus interrupted when the tissues that have already been compromised become necrotic. Alternatively, the production of JA and ethylene can lead to induced systemic resistance (ISR), which occurs upon exposure to non-pathogenic, root-associated bacteria and may be used against necrotrophic pathogens [41,44,45,46,47].

The activation of specific defenses relies on signals that function as amplifiers or regulators. The best-studied phosphorylation-dependent signal regulation cascades are calcium-dependent protein kinases and mitogen-activated protein kinases. A rapid, pre-transcriptional defense response is a product of these cascades. On the other hand, a long-term, large-scale, transcriptional response to stress is also initiated [48].

Pathogens can, nevertheless, manage to overcome the immune hurdles set up by plants. Colonization is then successful, and symptoms of the disease appear [49]. The bacterial repertoire of survival strategies includes resistance to antimicrobial compounds; efflux pumps that detoxify the cells; sporulation (in the case of gram-positive bacteria); the secretion of effector proteins; the synthesis of enzymes, toxins, and phytohormones; biofilm formation; the formation of persistent cells; the production of virulence factors; and genetic adaptation [50,51].

Furthermore, phytopathogenic bacteria have secretion systems through which pathogenic factors from their cytosol go directly into plant cells. These factors are involved in virulence, immune responses, host specificity, the obtention of resources, and alterations in physiology and cell function, among others [52]. The known secretion systems differ in structure, function, and specificity:

* Type I secretion systems (T1SS), which include an ABC (transmembrane) transporter, enable the transport of polypeptides such as metalloproteases, lipases, and toxins. They can be found, for example, in *Dickeya* spp. and *Pectobacterium* spp. [52,53,54].

* Type II secretion systems (T2SS) allow bacteria to translocate hydrolytic enzymes, toxins, etc. One of their main components is a cytoplasmic ATPase. They are present, for instance, in *Pectobacterium* spp., *Xanthomonas campestris*, and *Xanthomonas axonopodis* pv. *citri* [55,56].

* Type III secretion systems (T3SS) do not secrete but rather inject effector molecules (generally enzymes with proteolytic activity) into the host cell, which manipulate cellular activity at the convenience of the pathogen and repress the host’s immune response [57,58]. More precisely, these effectors can cause a hypersensitive response (HR) in resistant plants and pathogenesis in susceptible plants [59]. T3SS are present in the plasma membrane of many gram-negative bacteria such as *Pseudomonas syringae*, *Dickeya dadantii*, *X. campestris*, *Ralstonia solanacearum*, and *Erwinia* spp. [60,61], and their complex “injectosome” machinery is encoded by *hrp* genes Several lines of research are currently exploring the feasibility of interfering with effectors to control phytopathogenic disease. Some of the effectors under study are AvrPphEPto in *P. syringae*, which is involved in programmed cell death [62], and RipBJ and RS 1002 in *R. solanacearum*, which induce an HR [63,64].

* Type IV secretion systems (T4SS), whose machinery is not well known, are encoded by chromosomes or plasmids and transport virulence factors, proteins, and nucleic acids. Bacteria such as *Agrobacterium tumefaciens*, *Pseudomonas aeruginosa*, *Escherichia coli,* and *Xanthomonas* spp. have them [65,66,67].

* Type V secretion systems (T5SS), consisting of autotransporter proteins, are smaller than the others. Present in *Xanthomonas* spp., they are involved in adherence to host surfaces, colonization, invasion, and biofilm formation [68,69].

* Type VI secretion systems (T6SS), within the cytoplasmic membrane, transport molecules by perforation (in a similar manner to the mechanism in bacteriophages). They can be found in *Xanthomonas* spp., *P. syringae*, and *R. solanacearum* [70,71,72].

*Type VII secretion systems (T7SS), whose machinery has also been scarcely explored, have been described in some gram-positive bacteria [71].

*Type VIII secretion systems (T8SS) are made up of fibrous structures known as “curli”. They are responsible for adhesion, secretion, aggregation, and biofilm formation, and, therefore, sometimes related to colonization. They also transport amyloidogenic proteins. They have been described in *E. coli*, but not in phytopathogenic bacteria [73].

* Type IX secretion systems (T9SS), described in gram-negative bacteria, transport molecules across the membrane and are associated with adhesins [74].

Among all the defensive strategies that evolved for bacterial survival in unfavorable and changing environments, biofilm formation is perhaps one of the most important. It not only protects bacteria from adverse conditions but also causes significant biotic stress in plants. For this reason, the rest of this review will focus on biofilm formation by phytopathogenic bacteria that have a major negative impact on agriculture.

## 4. Biofilm: Composition, Functions, and Stages of Formation

Biofilm has garnered significant scientific interest in the last decade since 99% of the bacterial population can produce it at some point in their life cycle [75]. Biofilms are microbial communities inside structures of their own making, which can adhere to living or inert surfaces. About 10–25% of a biofilm consists of bacterial cells that can belong to members of the same species (in which case the biofilm is simple) or different species (mixed biofilm). The remaining 75–90% is made up of extracellular polymeric substances (EPS), which stabilize and give shape to the matrix [75,76]. This matrix confers enhanced protection to the cells within it against phagocytosis, harmful environmental conditions (pH, lack of nutrients, and mechanical forces), and antibiotics or antimicrobials: in fact, bacteria within a biofilm may be a thousand times more resistant to these agents [75,76,77]. Channels engineered on the inside, moreover, facilitate the circulation and exchange of water, nutrients, and enzymes as well as greater metabolic cooperation between members and the elimination of toxic metabolites [78].

Biofilm formation is a rapid, complex, and dynamic process that depends on changes in the cellular phenotype [79]. Its progressive stages may be summarized as follows:Adhesion: the microorganisms engage in weak interactions (acid-base, hydrophobic, Van der Waals, and electrostatic forces) to reversibly adhere to a surface.Colonization: irreversible bonds come about through hydrophilic/hydrophobic interactions; the bacteria use flagella, pili, and collagen-binding adhesive proteins.Development: EPS are secreted and there is a continuous proliferation and accumulation of cells.Maturation: the three-dimensional structure settles into its stable form featuring circulation and signaling channels.Active dispersal: groups of microorganisms separate from the mature biofilm to colonize new surfaces [78,80].

Certain bacteria within an established biofilm can evolve into persistent cells. These are genetically similar but physiologically different from the parent or primary cells (those that originally colonized the surface). Persistent cells are important in terms of resistance; their metabolism is inert, their replication is slow, and they regulate DNA-repair systems and antitoxin systems [81].

Since biofilm formation is cooperative, it would not be possible without quorum sensing (QS). This bacterium-to-bacterium communication system regulates not only the production of toxins, enzymes, biofilms, and EPS but also virulence factors and infectious processes in pathogenic bacteria [82,83] takes place through the synthesis of low molecular weight signaling molecules known as autoinducers (AI), which act as indicators of population density. When the concentration of these molecules exceeds a certain threshold, they are sensed by receptor molecules that are also synthesized by bacteria, and specific genes are activated (such as those responsible for biofilm production). In short, QS helps bacteria organize themselves into a community through a unified response and, thus, enhances their chances of survival [84].

The regulation of this communication system is very complex and varies from one bacterial species to another. Some of the molecules known to be involved are acyl-homoserine lactones (AHL) in gram-negative bacteria, autoinducing peptides (AIP) in gram-positive bacteria, and autoinducers 2 and 3 (AI-2 and AI-3) in both [79,83,85].

More specifically, QS in most gram-negative bacteria is regulated by a LuxI-LuxR-type system that becomes transcriptionally activated at a certain concentration of extracellularly diffused AHLs. Genes associated with biofilm formation are subsequently expressed. Some gram negatives, such as *X. campestris* and *R. solanacearum*, can also synthesize a diffusible signal factor (DSF) as an AI [86].

In gram-positive bacteria, it is a small oligopeptide (a mature AIP) that is produced and then expelled from the cell. Increasing concentrations of this peptide allow it to bind to a histidine kinase enzyme. A phosphorylation cascade ensues and genes related to biofilm formation are expressed [87].

In phytopathogenic bacteria, a global second messenger called cyclic di-GMP (cyclic guanosine monophosphate, abbreviated as cGMP or c-di-GMP) is involved in biofilm formation as well [88]: it regulates EPS biosynthesis and the transition from a mobile planktonic state to one of aggregation. Other processes regulated by this molecule are virulence, the cell cycle, and cell differentiation [82].

## 5. Social Behavior of the Bacterial Population in the Biofilm Matrix and Its Relationship with Pathogenicity

Social interactions, a common feature within the prokaryotic world [89], make it possible for bacterial populations to respond to environmental variations dynamically and collectively [90]. This adaptive behavior depends on changes at the level of gene expression, which are a result of chemical information in the form of diffusible signal molecules being produced and detected through QS (see Section 4). Moreover, many bacterial species that live in association with plants do so as members of polymicrobial communities, which are self-organized into highly complex biological structures [75,91].

These concepts are the basis of sociomicrobiology, i.e., the study of microbial social behavior. Seen from this perspective, bacteria within a group can act in their self-interest, cooperatively, or with altruism. They may compete against each other, divide labor amongst themselves, or function as donors/recipients in the transfer of genetic material. The biofilm matrix particularly accentuates cooperation and competition. The former occurs when the amount of biofilm formed by an individual species is less than the sum of all the biofilms formed by bacteria from the same environment. The latter is due to the amount of biofilm by one species surpassing the total production by bacteria from different environments [91,92].

In nature, most microorganisms are part of biofilms established as multispecies consortia. This makes sense, given that mixed biofilms tend to have a larger biomass and be more resistant [93]. The dynamics inside these biofilms are probably finely regulated through intra- and inter-species signaling, and members unable to synthesize QS molecules may effectively detect those produced by others [75,94].

Phytopathogenic bacteria within a biofilm can thus act in a coordinated manner to survive, outcompete other microbes, persist in nature, colonize host plants, and eventually infect them. All these processes are far more challenging for bacteria attempting them individually. Biofilms, then, contribute significantly to pathogenicity. In fact, some pathogenic microorganisms can only regulate and express their virulence if environmental conditions and cell density are optimal. Put otherwise, their pathogenicity relies on QS-mediated social activity. For example, *P. syringae* pv. *actinidiae*, the causative agent of bacterial canker in kiwi (see also 7.2.), colonizes the plant phyllosphere saprophytically and only penetrates the plant through wounds and natural openings when bacterial cell density is suitable [95]. Therefore, the study of these ecological relationships is a topic of increasing interest for the understanding of plant pathologies [96,97].

## 6. Biofilm in Plants

Biofilms were first observed on leaf surfaces in the early 1960s and on roots in the 1970s. Since then, evidence has accumulated of plant-associated bacterial aggregates, microcolonies, and biofilms. Plants harbor a diverse bacterial community on leaves, roots, shoots, and/or within their tissues. Bacteria can even be found in the depressions at the junctions of epidermal cells.

Plant-bacteria interactions can either harm or benefit plant health and yield. In other words, some biofilm-forming bacteria boost plant growth and protect against disease, while others are phytopathogenic [82,98,99]. For example, plant-growth-promoting rhizobacteria (PGPR) ensure effective root colonization by forming a biofilm, and in so doing enhance the plants’ ability to synthesize useful hormones and acquire nutrients, their resistance to stress, and, therefore, their yield [100,101]. In contrast, the biofilms formed in vascular systems by pathogenic bacteria, such as *R. solanacearum*, *Xyllela fastidiosa*, and *Pantoea stewartii*, cause wilting [98,102]. Certain human biofilm-producing pathogens, such as *P. aeruginosa*, can also infect the roots of plants, such as *Arabidopsis*, by producing the same virulence factors [102].

The ability of a pathogenic bacterium to successfully produce a biofilm on a plant host is strongly influenced by plant-microorganism interactions, the environment, and the host’s own immune response, physiological status, nutritional status, and signaling system, as described in Section 5 [82].

## 7. Biofilm Formed by Phytopathogenic Bacteria

The impact of phytopathogenic biofilms on agriculture cannot be understated. Formed on leaves (mesophyll, parenchyma), in the rhizosphere, and/or in vascular bundles [103,104], they reduce crop yield and quality and affect the safety of agricultural products intended for human consumption and animal feeding. In addition, the EPSs secreted by phytopathogenic bacteria interfere with the proper functioning of plant tissues and organs [103]. All of this seriously undermines food security at a time when booming populations around the world require high productivity rates, as indicated by the UN [100].

The current strategies to control biofilms include pesticides and antibiotics, but they are not efficient enough and they pose their own risks. Infections usually reappear and the overuse of chemical products causes water and soil contamination. If new management strategies are to emerge, they will demand thorough knowledge of how phytopathogenic biofilm is produced and regulated [85].

Several studies have focused on the importance of biofilm formation in plant pathogenesis. Many phytopathogenic bacteria produce biofilm on the leaf surface, such as *P. syringae* pv. *theae*, which is able to survive drought by living within biofilms on tea leaves [105]. Other *Pseudomonas* spp., such as *P. aeruginosa* on *Arabidopsis taliana* and *P. syringae* pv. *syringae* B728a, form biofilms on trichomes. These structures retain water and contain nutrients, two key components for the creation and endurance of the 3-D matrix [73]. Biofilm may be formed in xylem vessels and roots as well (see following subsections). Other examples of biofilm-producing phytopathogenic bacteria are *A. tumefaciens*, *Xylella fastidiosa*, *Erwinia amylovora*, *P. stewartii*, *R. solanacearum*, and *X. campestris* [74,82].

The disease cycle, which comprises differentiated stages and is represented in Figure 1, starts with a source of inoculum and ends with the appearance of symptoms such as tumors [106], decay and chlorosis [107], blight [108], wilting [109], rot [110], and cankers [111].

QS is responsible for regulating pathogenicity and colonization [83,112], and some phytopathogenic bacteria may have more than one QS system (featuring AHLs or diffusible signal factors, DSF) as well as a virulence factor modulation system [113]. For bacteria such as *E. amylovora*, *P. syringae*, *Xanthomonas* spp., and *Ralstonia* spp., T3SS are also involved in pathogenesis since they enable the direct introduction of pathogenic proteins into host cells. These systems are encoded by hypersensitivity response and pathogenicity *(hrp*) genes, which are classified into two groups with different organizations and modes of regulation. The *hrp* in group 1, typical of *Pseudomonas*, *Erwinia,* and *Pantoea spp*., are activated by complex regulatory pathways that end in proteins HrpS and HrpL. *Xanthomonas* and *Ralstonia* spp., on the other hand, have group II *hrp* [114].

### 7.1. Phytopathogenic Bacteria that Colonize Xylem Vessels

The causative agent of fire blight, *E. amylovora*, colonizes rosaceous plants by regulating their immune responses and physiology through a T3SS. In addition, it produces two EPS, amylovoran and levan, to form a biofilm within the vascular tissue. The synthesis of these polymers and that of cellulose, another component of the biofilm matrix, are positively regulated by an increase in intracellular c-di-GMP [82,115].

Gram-negative bacteria belonging to the genus *Dickeya* (formerly *Erwinia*) cause soft rot by synthesizing pectinase, an enzyme that degrades pectin in the cell wall and the middle lamella. The process is known to be regulated by QS and virulence factor modulation [116]. *D. dadantii*, responsible for stem and root rot in sweet potatoes, grows in biofilms and regulates the colonization of intercellular spaces and xylem vessels through flagella-mediated motility, the synthesis of LPS and extracellular polysaccharides, biofilm formation, and a T3SS [117,118]. Another example within the same genus is *D. zeae,* which affects economically important crops such as maize, bananas, rice, and potatoes [119].

*Pectobacterium carovotum* (also formerly classified as an *Erwinia*, but now within the genus *Pectobacterium*) forms a biofilm in the xylems of susceptible host plants, such as potatoes [82]. Different strains vary in their virulence depending on the concentrations of AHLs, which once again shows how crucial QS is to successful infection. Much like *Dickeya* spp., *P. carovotum* hydrolyzes pectin between plant cells and brings about soft rot. In addition, it can “hijack” the host’s genes to foster the development of disease [120,121].

Pierce’s disease in vines and variegated chlorosis in citrus plants occur when extensive biofilms of *X. fastidiosa* are created in the vascular system. In those areas where biofilm manages to block the nutrient flow, visible symptoms appear. Grasshoppers act as vectors: the bacterium colonizes their large intestine thanks to a QS system whose signal molecules are DSF and *rpf* gene products. When *X. fastidiosa* senses that its population density within the plant is high, the synthesis of c-di-GMP is inhibited and the formation of EPS and biofilm in the xylem are promoted. Conversely, low density is associated with an increase in intracellular c-di-GMP and the inhibition of adhesion and biofilm formation. This makes it possible for the pathogen to circulate freely through the plant and to be transferred into the insect vectors that feed on the plant’s sap [82,102,122,123].

*X. campestris* pv. *campestris*, the causal agent of black rot in crucifers, colonizes the xylem after gaining access to the plant through wounds or hydathodes. It synthesizes xanthan gum to form biofilm and degrading exoenzymes that promote virulence. Its aggregation is regulated by c-di-GMP and a two-component RpfC/RpfG system, in which RpfC is the histidine kinase sensor and RpfG is the response-regulating protein. When intracellular c-di-GMP is high, a protein similar to the cyclic AMP Clp receptor changes its conformation and cannot bind to target sites such as the promoter region in *manA*, which codes for endomannanase, a biofilm-dispersing enzyme. Genes in the *xag* cluster, on the other hand, code for a glycosyl transferase that is important for biofilm formation [82,122].

Sweet corn and maize may suffer from Stewart’s wilt, a disease transmitted by the corn flea beetle and caused by *P. stewartii* subsp. *stewartii*. Through the intervention of an *hrp*-encoded Hrp T3SS and the effector WtsE, the bacterium infects the apoplast and the xylem. There, its population density grows and dense biofilms are formed, encapsulated in a slime exopolysaccharide called stewartan. The water flow is blocked and symptoms appear, ranging from chlorotic lesions on leaves that eventually become necrotic and delay growth to rapid wilting and death in more susceptible plants. Stewartan also facilitates the pathogen’s movement through the vessels or intercellular spaces, which increases virulence [114,122,124,125].

Fatal wilt in more than 200 plant species is caused by several *R. solanacearum* strains that live in the soil, preferably in the deeper layers. Water and infected weeds can function as its reservoirs. It infects the roots and travels to the xylem where it multiplies, obstructs the vessels with large amounts of EPS, and, in the end, causes the plant to wither and die. Other factors implicated in its virulence are the synthesis of wall-degrading enzymes [122,126,127], chemotaxis, and motility. Chemotaxis (as described in Section 3) takes place when receptors in the bacterial cell membrane detect specific chemical substances to which they are attracted. Motility can take different forms depending on the appendages driving it. A rotary movement called swimming is produced through the use of polar flagella, while coordinated multicellular movement can be achieved by extending, attaching, and retracting type IV pili (see Section 4) [128,129].

A gram-positive pathogen that targets xylem vessels is *Clavibacter michiganensis*. *Clavibacter michiganensis* subspecies *michiganensis* infects tomatoes, and *Clavibacter michiganensis* subspecies *sepedonicus* gives rise to ring rot in potatoes [122,130,131]. *C. michiganensis* subsp. *michiganensis* is the causal agent of bacterial wilt and canker in tomatoes. Unilateral wilting in the host plant during the early stages of infection may mean that the pathogen has invaded the protoxylem, not the adjacent vessels, in which case the plant is still safe as a nutritional source. Dehydration and death ensue when *C. michiganensis* multiplies and produces EPS and glycoproteins to create large biofilms that decrease the water flow [132,133,134].

### 7.2. Phytopathogenic Bacteria that Colonize Root Tissues

Dicotyledonous plants suffer serious damage when tumors called “galls” appear in the junction between the roots and the stem. The formation of these tumors is induced by genes inside a Ti plasmid that belongs to a parasitic bacterium, *A. tumefaciens,* which produces biofilm mainly on the roots. For this purpose, it synthesizes cellulose and a unipolar polysaccharide adhesin (UPP) when intracellular c-di-GMP is high [75,82,135]. Some of the regulatory pathways for biofilm production in *A. tumefaciens* include an oxygen limitation response pathway, a two-component PhoR-PhoP system (involved in adhesion and biomass increase), and a regulator of ExoR secretion (which has to do with motility) [136].

A soil-borne pathogen that has been reclassified as *A. rhizogenes* (after being considered a rhizobium) forms large biofilms and causes hairy root disease in hydroponics. It has a signaling system made up of AHLs, and it introduces its DNA into the plant genome to live at the expense of its host [137,138,139].

### 7.3. Phytopathogenic Bacteria that Colonize Parenchymal Tissues

Phytopathogenic bacteria that form biofilm on leaf surfaces enter the leaf through natural openings called stomata. Many of them synthesize an ice nucleation protein that allows them to survive inside the leaves at low temperatures [140]. *Pseudomonas* spp. are prominent examples. Their ability to form a biofilm has been shown to depend on the incubation time and the availability of nutrients since different species adopt different strategies for colonization [141]. *P. syringae,* a widespread pathogen among crops, poses a significant threat to food security worldwide. More than 50 pathovars have been identified; each one with a high degree of host specificity and, therefore, the ability to infect only a limited number of plant species or even a few cultivars of a single species. This specificity is the basis for classifying *P. syringae* strains into different pathovars (pv.) [142,143]. For instance, the *Pseudomonas* strain that infects tomato plants has been named *P. syringae* pv. *tomato*, and the one responsible for canker in kiwifruit is *P. syringae* pv. *actinidiae*. Their production of biofilm on the leaf surface and their virulence are favored by the production of alginate, a polysaccharide [144,145]. This production and that of acetylated cellulose, another important component of their biofilm matrix, is regulated by *algU* genes, which are additionally involved in osmotolerance and motility. The synthesis of acetylated cellulose is also regulated by a wssABCDEFGHI operon. As with other bacteria reviewed here, c-di-GMP plays its part in biofilm formation [61,146].

*P. syringae* strains, moreover, use a T3SS to inject plant cells with virulence factors, such as protein effectors and a phytotoxin (coronatin) that mimics the plant hormone methyl jasmonate. This grants bacteria the ability to decrease the host’s immune response and increase its susceptibility to disease [82,147,148]. The T3SS is encoded by a group of *hrp* and hypersensitive and conserved response (*hrc*) genes, which are strictly controlled by the enhancer-coding proteins HrpR and HrpS. These proteins cooperatively activate HrpL expression, controlled by sigma-54. The emergence of *P. syringae* strains that have developed resistance to traditionally used antimicrobials makes their control all the more difficult [149,150]. *P. savastanoi* pv. *glycinea*, another *Pseudomonas*, also uses an *hrp*-encoded T3SS, coronatin, and biofilm to cause blight in soybean [148,151]. An example outside the *Pseudomonas* genus (although it was formerly classified within it) is *Acidovorax citrulli,* the causative agent of fruit spots in cucurbits. Its pathogenicity is not only related to biofilm formation but also to a T6SS, an R3SS, and the use of QS [152,153]. The involvement of biofilm in phytopathogenic disease has been depicted in Figure 2. Table 1 details biofilm-forming phytopathogenic bacteria which have been reported since 2015, including the diseases and symptoms with which they are associated.

## 8. Conclusions and Perspectives

After outlining the primary sources of abiotic stress in plants, this review summarized relevant aspects of biotic stress caused by the main phytopathogenic bacteria, with an emphasis on the advantages conferred by the ability to form biofilms. Phytopathogenic bacteria within biofilm have higher virulence and pathogenicity, and they can resist antimicrobials synthesized by the host and overcome other plant immune responses. The fact that biofilms can be constantly remodeled and restructured also makes them versatile systems in the face of changing environments in terms of UV radiation, desiccation, and nutrient availability, among many other factors. This enhanced survival in the plant (whether in or on leaves, roots, conductive vessels, etc.) as well as in the environment makes phytopathogens all the more harmful to crop yield, health, and quality.

An understanding of the nature and role of biofilm in pathogenesis, therefore, is crucial to effectively control and/or minimize diseases caused by biofilm-producing plant pathogenic bacteria. Currently, much is known about biofilm structure, EPS composition, signaling mechanisms, plant penetration and colonization, and symptoms of disease. These variables depend mainly on the microbial species, its metabolic activity, nutrient availability, the ecosystem, and the plant’s growth stage. The bacterial communication process is also very well characterized. Nevertheless, our understanding of bacterial interactions in a variety of microenvironments, as well as of biofilm formation and the sophisticated mechanisms that regulate plant-host associations, could benefit from interdisciplinary and in vivo studies. Every new piece of evidence on the extremely complex strategies in which bacteria engage for survival and infection will contribute towards the development of more effective and environmentally-friendly schemes for disease management.

## Figures and Tables

**Figure 1 plants-12-02207-f001:**
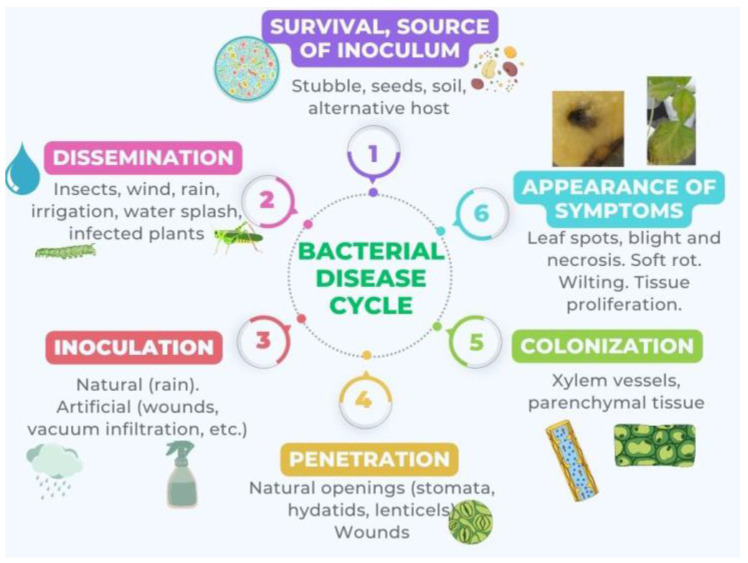
Cyclic model of disease by phytopathogenic bacteria consisting of several distinct stages: 1. Survival or source of inoculum: The bacteria survive outside the host by developing in another environment or by remaining dormant. 2. Dissemination: The bacteria spread. 3. Inoculation: The bacteria come into contact with the plant. 4. Penetration: The bacteria enter the plant. 5. Colonization: The bacteria disseminate within the plant. 6. Appearance of symptoms: This is the result of bacteria producing phytotoxins, EPS, exoenzymes, phytohormones, etc.

**Figure 2 plants-12-02207-f002:**
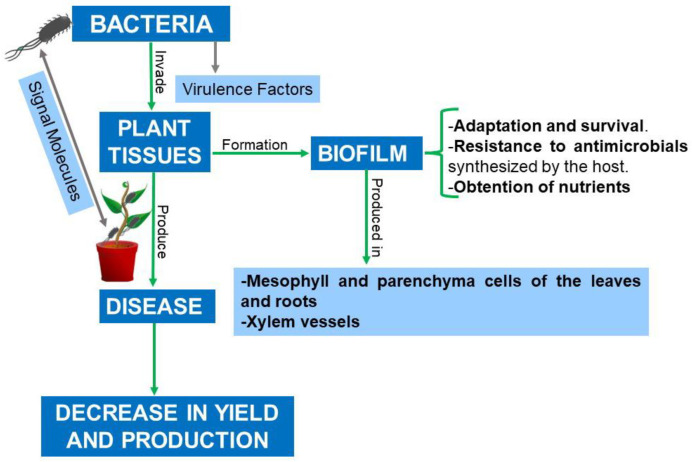
Involvement of biofilm in interactions between plants and phytopathogenic bacteria.

**Table 1 plants-12-02207-t001:** Summary of biofilm-forming phytopathogenic bacteria which have been reported since 2015, including the diseases and symptoms to which they are associated.

Species (Infectious Agent)		Host (Sensitive Plant)	Disease	Symptoms	References
** *Pseudomonas syringae* **	pv. *syringae*	Wheat	Bacterial leaf blight	Water-soaked lesions that turn gray-green, leaf necrosis	[154]
	pv. *syringae*	Tomato, cereals, citrus plants, kiwifruit	Spot, speck, and bacterial blight	Foliarand stem necrotic lesions; basal stem and root rot	[155]
	pv. *atrofaciens*	Wheat	Basal glume rot	Dull brownish-blackish discoloration on the lower part of the glume	[154]
	pv. *tomato*	Tomato	Bacterial speck	Chlorosis and necrotic lesions	[156]
	pv. *actinidiae*	Kiwi	Bacterial bleeding canker	Brown leaf spots with chlorotic haloes, fruit specks and scabs, brown discoloration of buds, and cankers with exudates on trunks and twigs	[157]
	pv. *phaseolicola*	Bean	Halo blight	Necrotic lesions on the leaf	[158]
*Pseudomonas savastanoi* pv. *glycinea*		Soybean	Bacterial leaf blight	Circular necrotic lesions on leaves surrounded by a chlorotic halo	[151,159]
*Xanthomonas citri* subsp. *Citri*		Citrus plants	Citrus canker	Erumpent lesions on fruit, foliage, and young stems	[160]
*Xanthomonas axonopodis* pv. *phaseoli*		Bean	Common bacterial blight	Dark green water-soaked lesions, necrotic symptoms on the margins of leaves	[161,162]
*Xanthomonas oryzae* pv. *oryzae*		Rice	Bacterial leaf blight	Tannish gray-white lesions along the veins	[51,163]
*Xanthomonas campestris* pv. *campestris*		Cruciferous plants	Black rot	V-shaped necrotic lesions on the foliar margins and blackened veins	[164]
*Xanthomonas translucens* pv. *undulosa*		Wheat	Bacterial streak and black chaff disease	Water-soaked necrotic streaks which eventually change into translucent lesions	[154]
*Pantoea stewartii* subsp. *stewartii*		Corn	Stewart’s wilt, severe seedling wilt	Water-soaked lesions, wilting in young seedlings	[165]
*Pectobacterium carotovorum* subsp. *carotovorum*		Potato	Soft rot	Severe bacterial tuber soft rot	[166]
*Pectobacterium carotovorum* subsp. *brasiliense*		Potato, tomato, cucumber, radish	Soft rot	Severe and typical bacterial soft rot, water-soaked and macerated tissues	[167,168]
*Agrobacterium tumefaciens*		Dicotyledonous plants	Crown gall	Tumors	[169]
*Rhizobium rhizogenes*		Tomato, cucumber, apple	Hairy root	Smaller root structures spring out at right angles from the main root. In the aerial form, tumors or knots (woolly-knots) appear in the limbs	[137,138]
*Clavibacter michiganensis*		Tomato	Bacterial canker, wilt disease	Deterioration of the internal vascular tissues, stem cankers, foliar chlorosis, unilateral wilt, marginal leaf necrosis, fruits with localized bird’s-eye spots	[131,170]
*Xylella fastidiosa*		Citrus plants grape, coffee, almond, olives, peach, blueberry, among others	Pierce’s disease, leaf scorch, and citrus-variegated chlorosis	Leaf chlorosis, marginal scorching, and/or dwarfing, depending on the host	[5,171]
*Erwinia amylovora*		Apple, pear	Fire blight	“Shepherd’s crook” of the twigs and a yellowish bacterial exudate on the infected tissues. Infection of leaves at shoot tips, wilting of leaves, cankers	[172,173]
*Ralstonia solanacearum*		Tomato, brinjal, tobacco, potato, banana	Bacterial wilt	Rolling of leaves, chlorosis, and necrosis	[109]
*Dickeya dadantii*		Sweet potato	Bacterial stem and root rot	Maceration of plant tissues	[117]
*Acidovorax citrulli*		Melon, watermelon, pumpkin	Bacterial fruit blotch	Water-soaked seedlings and light brown-reddish lesions on the leaves, small water-soaked regions on the fruit surface	[126,174]

## Data Availability

All the data are included in the manuscript.

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
