# Peer review of "Biofilm-Forming Ability of Phytopathogenic Bacteria: A Review of its Involvement in Plant Stress"

_plants, 2023, doi:10.3390/plants12112207_

Round 1
Reviewer 1 Report
This paper was difficult to follow and requires serious re-organization. It started off with a lengthy section on abiotic stress, tried so hard to fit in secretion systems, then dove-tailed into biofilms. It seems the authors wanted to write on Biofilm-forming plant pathogenic bacteria and the extent of the disease they cause. I would recommend a review of the title of the paper to reflect this. Section 6 talks about different biofilm forming bacteria and this seems to be the focus of the paper. Authors should focus on this and present more information.
Abstract
Not sure if it is necessary: Research on plant stress, whether abiotic or biotic, is currently booming, since crop
productivity must be guaranteed for a growing global population whose food demand is also on the rise.
Authors can start from the second sentence.
Introduction
General comment: The focus on phytopathogenic bacteria began at line 134. That’s a lot of good but unnecessary preamble for the focus topic.
Line 44: please review “is not always possible”. The example authors gave (drought vs fungi) does not fall into the realm of impossibility.
Line 65: please check the reference [11] for appropriateness. Same for [12].
Line 140: there are other ways of bacterial infection: seed transmission, insect vectors, etc.
Line 169-177: provide listed strategies in sentence, not as list.
Line 178: colloquial: reword “thanks to”. You meant “through which”?
Line 195: Bacteria such as Agrobacterium tumefaciens have them: please check other bacteria too for T4SS
Line 203: why skip T8SS?
Line 270: please revise, too colloquial: Certain human pathogens are behind disease in plants
as well, where they have the same virulence factors as in humans
Line 398: 50 strains: strains or pathovars?
Line 401: Poorly worded: pv. tomato is for all tomato pathovars, not just strain DC3000.
Overall, the writing reads fairly well. However, there were some use of colloquial and poorly worded languages in the manuscript.
Reviewer 2 Report
Carezzano et al., make a very good review of the several aspects involved in the pathogenicity of phytopathogenic bacteria and focus in the biofilm formation as a key virulence factor. The review is up-to-date and pertinent, and it is a good approach material for consultation by both advanced and beginning researchers. However there are a few aspect that need to be addressed :
1. lines 189-192. Protein secretion system III please mention the importance fo this secretion system in the transportation of some bacterial effectors responsible for pathogenicity in host cells and please include a few examples
2. Please in the item 4 "Biofilm: Composition, functions, and stages of formation " include the regulation aspect mainly based in the quorum sensing regulation .
3. finally, if it is possible to introduce a small section related to the social behaviour of bacterial population in the biofilm matrix and its relation with the pathogenicity .
The quality of the English is good enough , easy to read, however, a review by a native speaker is recommended.
Reviewer 3 Report
Dear Author, I reviewed the manuscript (plants-2375496) entitled Plant Stress: A Review Focused on Biofilm-Forming Pathogenic Bacteria. This manuscript presents relevant information about pathogenic bacteria biofilm development. However, some sections of the presented data can be improved. For this reason, I consider that this manuscript needs minor changes to be considered for publication in this journal.
Additional comments.
Highlight the advantages of studying different biofilm development of phytopathogenic bacteria.
Check paragraphs extension in this manuscript.
Try to include different quorum sensing systems bacteria use and their relation with biofilm development.
Include interactions with interspecies of different bacteria in biofilm development on field areas or crops.
Include future trends to keep working with the obtained data.
Try to conclude with a general statement of the most relevant part of this study.
Round 2
Reviewer 1 Report
The manuscript is very much improved. However, I have a question on the title: are there reported non-biofilm forming plant pathogenic bacteria? The authors could highlight this briefly in the paper to further add to the reason why they focused on biofilm forming bacteria. Otherwise, the title should be modified in general, and the word “focus” be removed in whatever form the authors decide to reword the title.
Line 89-94: what was the nature of salinity in both cases? Na+, Ca2+, K+ etc?
Line 195: TS33 to T3SS
Line 202: See this for T4SS in Xanthomonas: https://www.frontiersin.org/articles/10.3389/fmicb.2022.835647/full
Line 228: please use a different word than “boomed” and provide reference supporting the statement: Research on biofilm has boomed over the last decade, based on the notion that 99% of the bacterial population can produce it at some point in their life cycle.
Line 340: please revise: systems (rapid regulation by c-di-GMP, for instance) pathogenesis, physiological,…
Line 344-388: enough introduction has been given in preceding sections. At this point, authors should dive into the focus of each section. Please delete 344 to 348.
Line 353, 381: what is EPS
Line 408: correct Erwinya
Line 424: X. campestris pv. campestris please put a dot after pv throughout the paper.
Line 454: write out in full: Clavibacter michiganensis subspecies michiganensis; Clavibacter michiganensis subspecies sepedonicus.
Line 455-459: please be specific. Which of the diseases are authors describing?
Line 486-487: remove DC3000, DC3000is only a strain. It’s enough to say P. syringae pv tomato.
Minor edits required but generally good.
Reviewer 2 Report
The authors have made a substantial improvement to the manuscript, important aspects such as the regulation of biofilm formation by quorum sensing systems , bacterial social behaviour and the role of protein secretion system III in pathogenicity are now incorporated into the manuscript. This a great review and I am sure that outcomes in a highly consulted paper.
All my request were addressed by authors and I my opinion the manuscript in the currently state has the conditions to be published in plants
The quality of the English is quite good, it has had a notable improvement in this version, however, some minor aspects of punctuation should be revised
